# The Clinical Impact of Neoadjuvant Endocrine Treatment on Luminal-like Breast Cancers and Its Prognostic Significance: Results from a Single-Institution Prospective Cohort Study

Covadonga Martí [1,2,*], Laura Yébenes [1,3,4], José María Oliver [1,5], Elisa Moreno [1,2], Laura Frías [1,2], Alberto Berjón [1,3,4], Adolfo Loayza [1,2], Marcos Meléndez [1,2], María José Roca [1,5], Vicenta Córdoba [1,5], David Hardisson [1,3,4,6,7], María Ángeles Rodríguez [1] and José Ignacio Sánchez-Méndez [1,2,4,6]

1   Breast Cancer Unit, Hospital Universitario La Paz, 28046 Madrid, Spain;
    laura.yebenes@salud.madrid.org (L.Y.); josemaria.oliver@salud.madrid.org (J.M.O.);
    elisa.moreno@salud.madrid.org (E.M.); lfrias@salud.madrid.org (L.F.);
    alberto.berjon@salud.madrid.org (A.B.); adolfo.loayza@salud.madrid.org (A.L.);
    marcos.melendez@salud.madrid.org (M.M.); mariajose.roca@salud.madrid.org (M.J.R.);
    mvicenta.cordoba@salud.madrid.org (V.C.); david.hardisson@salud.madrid.org (D.H.);
    mrpaton@salud.madrid.org (M.Á.R.); joseignacio.sanchez@salud.madrid.org (J.I.S.-M.)
2   Department of Gynecology, Hospital Universitario La Paz, 28046 Madrid, Spain
3   Department of Pathology, Hospital Universitario La Paz, 28046 Madrid, Spain
4   IdiPaz—Instituto de Investigación La Paz, 28046 Madrid, Spain
5   Department of Radiology, Hospital Universitario La Paz, 28046 Madrid, Spain
6   Faculty of Medicine, Universidad Autónoma de Madrid, 28046 Madrid, Spain
7   Center for Biomedical Research in the Cancer Network (CIBERONC), 28029 Madrid, Spain
*   Correspondence: covadonga.marti@salud.madrid.org

**Abstract:** Purpose: Neoadjuvant endocrine treatment (NET) has become a useful tool for the downstaging of luminal-like breast cancers in postmenopausal patients. It enables us to increase breast-conserving surgery (BCS) rates, provides an opportunity for us to assess in vivo NET effectiveness, and allows us to study any biological changes that may act as valid biomarkers. The purpose of this study was to evaluate the safety and effectiveness of NET, and to assess the role of Ki67 proliferation rate changes as an indicator of endocrine responsiveness. Methods: From 2016 to 2020, a single-institution cohort of patients, treated with NET and further surgery, was evaluated. In patients with Ki67 $\geq$ 10%, a second core biopsy was performed after four weeks. Information regarding histopathological and clinical changes was gathered. Results: A total of 115 estrogen receptor-positive (ER+)/HER2-negative patients were included. The median treatment duration was 5.0 months (IQR: 2.0–6.0). The median maximum size in the surgical sample was 40% smaller than the pretreatment size measured by ultrasound ($p < 0.0001$). The median pretreatment Ki67 expression was 20.0% (IQR: 12.0–30.0), and was reduced to 5.0% (IQR: 1.8–10.0) after four weeks, and to 2.0% (IQR: 1.0–8.0) in the surgical sample ($p < 0.0001$). BCS was performed on 98 patients (85.2%). No pathological complete responses were recorded. A larger Ki67 fold change after four weeks was significantly related to a PEPI score of zero ($p < 0.002$). No differences were observed between luminal A- and B-like tumors, with regard to fold change and PEPI score. Conclusions: In our cohort, NET was proven to be effective for tumor size and Ki67 downstaging. This resulted in a higher rate of conservative surgery, aided in therapeutic decision making, provided prognostic information, and constituted a safe and well-tolerated approach.

**Keywords:** breast cancer; endocrine therapy; neoadjuvant; resistance

## 1. Introduction

Neoadjuvant treatment with chemotherapy (NCT) is a widely accepted approach for high-risk or locally advanced breast cancers that enables both tumor size and axilla

involvement to be downstaged, and it has been crucial for the development of new drugs and the study of biomarkers. However, NET has taken longer to implement. Traditionally, these therapies were relegated to frail or elderly patients who were not candidates for chemotherapy or surgery [1–3], and it is only in recent years that the results of multiple prospective trials have made NET a very attractive therapeutic option for postmenopausal patients with estrogen receptor-positive (ER+)/HER2- tumors [4–8]. It produces good clinical responses, thus increasing BCS rates, and it also offers a unique opportunity for research. Most international guidelines have already included NET among their validated treatments for postmenopausal women with ER+/HER2- breast cancer [9–11].

The key point when neoadjuvant treatments are to be used is to make an accurate assessment of the in vivo response to therapy. The changes observed on imaging tests (ultrasound, mammography, and MRI) provide guidance in both NCT and NET [12–15], but no accurate biomarker of response has yet been identified for NET. Nevertheless, the short-term assessment of Ki67 levels does seem to correlate closely with the degree of response. A Ki67 reduction may occur after only a few weeks from the initiation of treatment, indicating cell-cycle arrest, induced by therapy, resulting from endocrine responsiveness [16–19]. On the other hand, high Ki67 levels after NET would suggest innate tumor resistance. A Ki67 > 10% after two to four weeks of endocrine therapy has been suggested as a cutoff for the early identification of nonresponders with an increased risk of relapse, and it is currently considered by many authors in various trials as the necessary threshold for maintaining or withdrawing NET [19–21].

Residual cancer burden (RCB) and pathological complete response (pCR) are considered to be accurate surrogate markers of response after NCT, mainly in high-risk breast cancer patients [22]; however, low rates of pCR in luminal tumors, with both NCT and NET, do not correlate with worse outcomes. Ellis et al. developed the preoperative endocrine prognostic index (PEPI) score, a prognostic model that incorporates standard pathologic staging variables and "on-treatment" biomarker values [5]. Consequently, low PEPI scores are related to a low recurrence risk. Although randomized NET trials, with or without other drug combinations, are numerous, real-world data, reflecting clinical practice, are scarce in this field [23].

The aim of this study was to analyze a prospective cohort of patients in our institution who had been treated with NET and subsequently operated on, evaluating the adherence and treatment tolerance, as well as the effectiveness of NET in tumoral downstaging. Furthermore, we wanted to assess the role and clinical applicability of Ki67 changes as an indicator of endocrine responsiveness and, therefore, of NET maintenance.

## 2. Methods

This was a prospective cohort study that included ER+/HER2- breast cancer patients treated with NET and subsequently operated on at our institution, a tertiary university center. Institutional Review Board approval of the study protocol was obtained. Inclusion criteria were as follows: postmenopausal status; ER+/HER2- breast cancer diagnosis in core biopsy; tumor size > 10 mm; cN0-N2; treatment initiated between 2016 and 2020. Patients with stage IV disease, previous NCT, or who were unfit for surgery were excluded. An initial mammogram, together with breast and axillary ultrasound, were performed. Tumor size was determined according to its maximum diameter. Breast and/or axillary ultrasound were also used for follow-up. All patients were treated with letrozole 2.5 mg daily, and this was continued until BCS was feasible, or the maximum tumor reduction was achieved. In those patients with a Ki67 proliferation rate ≥ 10%, a second core biopsy was performed after four weeks (Figure 1). Informed consent was obtained from all individual participants included in the study.

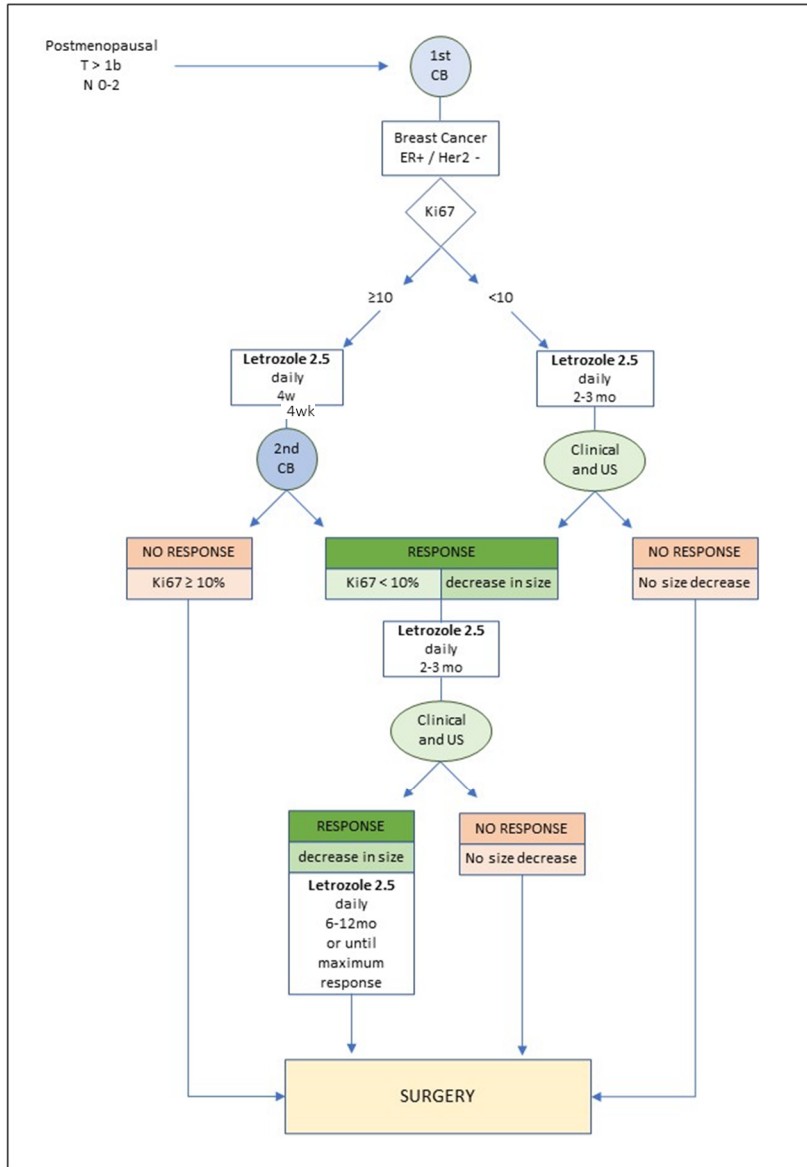

**Figure 1.** Study protocol. CB: core biopsy; wk: week; mo: month; US: ultrasound.

Data were collected from patient charts, including the following: age; histological type (ductal, lobular, other); histologic grade (G1, G2, or G3); ER; progesterone receptor (PR); Ki67 expression (%) (pretreatment, intermedial, and in the surgical sample); clinical and pathological size (mm); clinical node status (positive or negative); clinical stage (TNM) [24]; duration of treatment (months); surgical approach (lumpectomy or mastectomy); upfront axillary surgery (sentinel lymph node biopsy (SLNB), targeted axillary dissection (TAD), axillary lymph node dissection (ALND), or none).

Immunohistochemistry for ER (clone EP1, prediluted, Dako, Glostrup, Denmark) and PR (clone PgR 1294, prediluted, Dako) was performed according to international guidelines. Ki67 proliferation rate (clone MIB1, prediluted, Dako) was defined as the mean expression in the whole tumor area [25]. Negative HER2 status was confirmed by either immunohistochemistry (HercepTest, Dako), or by in situ fluorescent hybridization (IQFISH HER2, Dako) in equivocal cases. St. Gallen 2013 criteria for defining luminal A or B phenotypes were applied [26]. Changes in Ki67 proliferation rates were analyzed on the natural log-fold scale.

pCR was defined as absence of invasive disease in the breast and axilla (ypT0/is ypN0), and the PEPI score was calculated as described by Ellis et al. [5].

We conducted descriptive analyses on the data collected. Categorical variables were described as frequency or percentage. Continuous variables were reported as median and interquartile range (IQR). Comparison of characteristics among variables was performed using the most appropriate test. The type 1 error rate ($\alpha$) was set to 0.05. Statistical analyses were performed using SPSS v26 (IBM Corporation, Armonk, NY, USA). We used the STROBE cohort reporting guidelines [27].

## 3. Results

A total of 115 consecutive patients, who fulfilled the inclusion criteria, were included in the analysis. The characteristics of the cohort are described in Table 1.

**Table 1.** Patient, tumor, and treatment characteristics.

| Characteristics of the Cohort | *n* | (%) |
|---|---|---|
| | 115 | (100.0) |
| Clinical Stage | | |
| IA | 46 | (40.0) |
| IIA | 45 | (39.1) |
| IB | 0 | (0.0) |
| IIB | 19 | (16.6) |
| IIIA | 5 | (4.3) |
| Clinical node evaluation-cN | | |
| Negative | 94 | (81.7) |
| Positive | 21 | (18.3) |
| Histological type | | |
| Ductal | 85 | (73.9) |
| Lobular | 23 | (20.0) |
| Others | 7 | (6.1) |
| Histological grade | | |
| 1 | 24 | (20.9) |
| 2 | 77 | (66.9) |
| 3 | 14 | (12.2) |
| Immunophenotype | | |
| Luminal A-like | 48 | (41.7) |
| Luminal B-like | 67 | (58.3) |
| | Median | (IQR) |
| Age (years) | 69.0 | (62.0–78.0) |
| Pre-NET Size (mm) | 25.0 | (17.0–40.0) |
| Pre-NET ER expression (%) | 100.0 | (100.0–100.0) |
| Pre-NET PR expression (%) | 70.0 | (20.0–100.0) |
| Pre-NET Ki67 (%) | 20.0 | (12.0–30.0) |
| NET duration (months) | 5.0 | (2.0–6.0) |

IQR: interquartile range; ER: estrogen receptor; PR: progesterone receptor; NET: neoadjuvant endocrine therapy.

### 3.1. Clinical and Tumor Characteristics

The median age was 69.0 years old (IQR: 62.0–78.0), 69 (60.0%) patients were stage II–III, and 21 (18.3%) patients were cN1. Invasive ductal carcinoma comprised 73.9% (*n* = 85) of the cases, while 20.0% (*n* = 23) of the cases were invasive lobular carcinomas. The tumor grade was moderately differentiated (G2) in 67.0% (*n* = 77) of the cases, and poorly differentiated (G3) in 12.2% (*n* = 14) of the cases. The median pretreatment size was 25 mm (IQR: 17.0–40.0). All the tumors had 100% ER expression, while the mean PR expression was 70% (IQR: 20.0–100.0). The median Ki67 proliferation rate was 20% (IQR: 12–30), and 58.3% (*n* = 67) of the cases were luminal B-like.

### 3.2. Treatment Characteristics

The median duration of the treatment with letrozole 2.5 mg daily was 5 months (IQR: 2–6). No clinical progression or treatment abandonment was reported.

BCS was feasible in 98 patients (85.2%). An SLNB was performed on 86.9% (*n* = 100) of the patients. On all but two cN0 patients (both elderly and with comorbidities), an SLNB was carried out (*n* = 92). The sentinel lymph node (SN) was negative in 70 patients, while micrometastasis was described in 9 cases. In 12 out of 13 cases with a positive SN, Z0011 criteria were considered, and an ALND was avoided. An upfront ALND was carried out on 13 (11.3%) of the 21 cN1 patients. As part of another study conducted at our facility, an SLNB, combined with a TAD, was performed on eight women with low axillary burden. A further ALND was completed on three of these patients. Surgical management is summarized in Table 2 and axillary management is summarized in Figure 2.

**Table 2.** Pathological outcomes after NET.

|  | *n* | % |
|---|---|---|
|  | 115 | (100%) |
| Pathological complete response | 0 | (0.0) |
| Pathological node status |  |  |
| ypN0 | 71 | (61.7) |
| ypN1 | 44 | (38.3) |
| PEPI score |  |  |
| 0 | 53 | (46.1) |
| 1 | 17 | (14.8) |
| 2 | 8 | (7.0) |
| 3 | 23 | (20.0) |
| 4 | 10 | (8.7) |
| 5 | 2 | (1.7) |
| 6 | 2 | (1.7) |

PEPI: preoperative endocrine prognostic index.

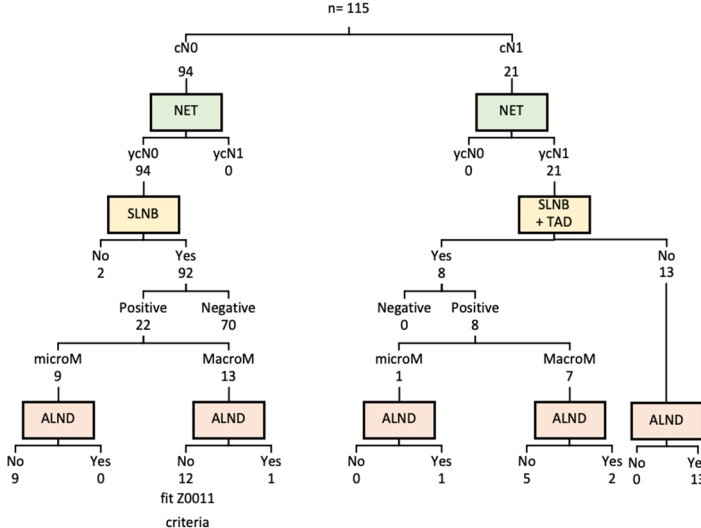

**Figure 2.** Axillary management. NET: neoadjuvant endocrine therapy; SLNB: sentinel lymph node biopsy; TAD: targeted axillary dissection; ALND: axillary lymph node dissection.

Adjuvant radiotherapy was used in 101 patients (87.8%), and chemotherapy was indicated for 29 patients (25.2%).

### 3.3. Pathological Changes after NET

Table 3 resumes the pathological and dynamic changes after NET. No cases of pCR were recorded. Fifty-three (46.1%) patients had a PEPI score of 0, and 23 (20.0%) had a PEPI score of 3. Forty-four patients (38.3%) were pN1, which indicates a false-negative radiological detection rate (FNR) of 24.5%. Significant downgrading of tumor size, Ki67, PR expression, and histologic grade was observed. After four weeks of treatment, an intermediate biopsy was performed on 78 patients (67.8%), which showed early changes in all these parameters. The reduction in Ki67 proliferation rates occurred most dramatically in the first four weeks, while, thereafter, their numbers tended to plateau. In two of the cases (2.6%) where an intermediate biopsy was carried out, an increase in Ki67 was reported. In these patients, NET was interrupted and surgery was indicated. The tumor size decreased from an initial median of 25 mm (IQR: 17–40) to a final median value of 15 mm (IQR: 10–20), implying a 40% reduction. Histological downgrading was reported in 31 out of 91 patients with grade 2 and 3 tumors (34.1%). The initial mean PR expression was 70% (IQR: 20–100), falling to 4% (IQR: 0–35) in the intermediate biopsy, and to 1% (IQR: 0–20) in the surgical sample. No differences in Ki67 fold change or PEPI score were observed between luminal A-like or B-like tumors (Figure 3a). A larger Ki67 fold change after four weeks was significantly related to a PEPI score of zero (Figure 3b).

**Table 3.** Biological changes due to NET.

| Header | | Pre-NET | | Intermedial | | Surgical Sample | |
|---|---|---|---|---|---|---|---|
| | | 115 | | 78 | | 115 | |
| | | median | I.Q.R | median | I.Q.R | median | I.Q.R |
| Size (mm) | | 25.0 | (17.0–40.0) | 19.5 | (13.0–30.0) | 15.0 | (10.0–20.0) |
| p | Pre-NET/Surgical<br>Pre-NET/Intermed<br>Intermed/Surgical | | | <0.0001<br><0.0001<br><0.0001 | | | |
| ER Expression (%) | | 100.0 | (100.0–100.0) | 100.0 | (100.0–100.0) | 100.0 | (100.0–100.0) |
| p | Pre-NET/Surgical<br>Pre-NET/Intermed<br>Intermed/Surgical | | | <0.05<br>ns<br>ns | | | |
| PR Expression (%) | | 70.0 | (20.0–100.0) | 4.0 | (0.0–35.0) | 1.0 | (0.0–20.0) |
| p | Pre-NET/Surgical<br>Pre-NET/Intermed<br>Intermed/Surgical | | | <0.0001<br><0.0001<br><0.005 | | | |
| Ki67 (%) | | 20.0 | (12.0–30.0) | 5.0 | (1.8–10.0) | 2.0 | (1.0–8.0) |
| p | Pre-NET/Surgical<br>Pre-NET/Intermed<br>Intermed/Surgical | | | <0.0001<br><0.0001<br>ns | | | |
| Histological Grade | | n | % | n | % | n | % |
| | G1<br>G2<br>G3 | 24<br>77<br>14 | (20.9)<br>(67.0)<br>(12.2) | 26<br>49<br>3 | (33.3)<br>(62.7)<br>(4.0) | 55<br>53<br>7 | (47.8)<br>(46.1)<br>(6.1) |
| p | Pre-NET/Surgical<br>Pre-NET/Intermed<br>Intermed/Surgical | | | <0.0001<br><0.0001<br><0.0001 | | | |

IQR: inter-quartile range; ER: estrogen receptor; PR: progesterone receptor; NET: neoadjuvant endocrine therapy; intermedial refers to data from 4-week biopsy.

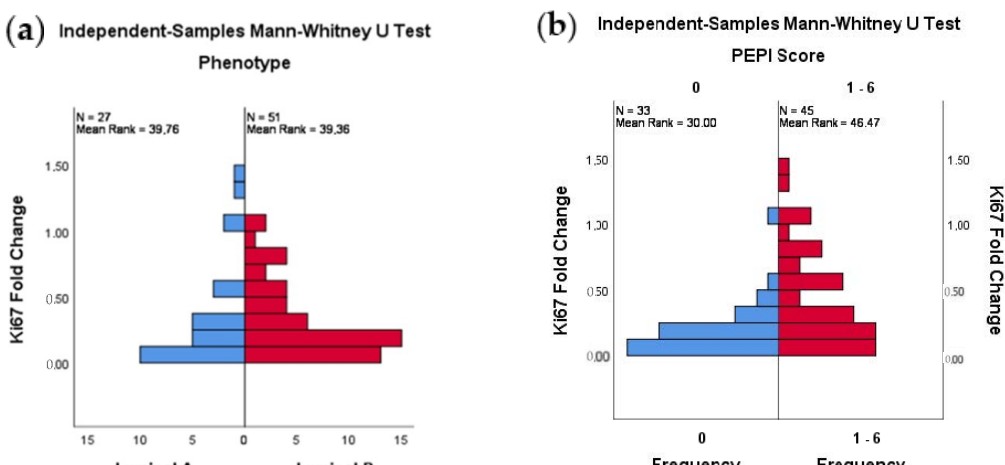

**Figure 3.** (**a**) Phenotype and 4wk-Ki67 fold change. (**b**) PEPI score and 4wk-Ki67 fold change.

## 4. Discussion

One of the main issues when using NET is the appropriate selection of patients. As is the case with endocrine adjuvant therapy (ET), NET should be considered for luminal-like tumors; however, although ET is recommended for tumors with even low ER levels [10,26], NET is normally used only with high ER expression tumors, since most trials have achieved better results in these cases [28,29]. In our cohort, all 115 tumors had diffuse ER expression, which could explain the high response rate we found. PR expression and/or Ki67 levels can also influence NET outcomes [30]. As a result, many authors only recommend NET for luminal A-like tumors, because initially low PR expression or high Ki67 levels (i.e., luminal B-like features) have been associated with a poorer response [29,31]. However, in our study, we included both luminal A- and luminal B-like tumors, and we did not find significant differences between them, with regard to clinical biological response or BCS rates, which could mean that NET may be a good approach, even in those cases. HER2 overexpression or a high histologic grade usually discourages the use of NET [29,31]. Although a few studies (IMPACT, P024) [4,6] included small subpopulations of HER2 positive, reporting over 50% of the clinical responses, we excluded ER+/HER2+ patients from our study. A higher histologic grade is considered a poor prognostic factor; consequently, many of these tumors are eligible for chemotherapy. Nevertheless, in our study, we observed a decrease in Ki67 levels, as well as in size, in most of these G3 cases. Additionally, up to 80% of these tumors changed to an intermediate grade after four weeks of treatment, although some of them were classified, again, as G3 in the surgical sample. It is difficult to know why a few of these tumors initially modify their histologic grade and then later go back to a higher grade, although sampling bias or a limited amount of tissue in the core biopsy could explain these findings. In any case, all these observations could imply that these types of tumors, in well-selected cases, may benefit from NET.

NET duration is usually established at 3 to 6 months, based on the regular duration of NCT, although the optimal period for this therapy has not yet been clearly determined. Several trials have been conducted to establish an optimal duration [32–34]. Most of them found that prolongation of NET produced extra benefits, such as higher pCR rates or axillary downstaging [35,36], although good responses and BCS rates can be achieved with an NET duration of six months or less [3]. The mean duration in our cohort was five months, becoming longer as we became more comfortable with NET implementation, and, at all times, it was conditioned in light of the response, either a decrease in Ki67 proliferation rates or clinical radiological changes.

As mentioned above, NET provides an opportunity for assessing the in vivo response of ER+ breast cancers. In our cohort, significant downgrading changes were observed, with regard to Ki67 levels, size, histological grade, and PR expression. Ki67 is a nuclear antigen expressed in proliferating tissues, and is considered a surrogate marker of cell proliferation.

A reduction in the Ki67 proliferation rate was identified only a few days after the initiation of NET [6,19]. We found a significant decline in Ki67 expression after four weeks, and this was usually maintained throughout the treatment; although, in a few cases, its value increased. The changes that appear in the Ki67 proliferation rate with NET constitute a biomarker with prognostic and predictive power, much more precisely related to long-term results than baseline Ki67 [17,19,20,25]. As demonstrated in the POETIC trial, patients with initially low Ki67 levels, or low postaromatase inhibitor-induced- Ki67 proliferation rates, probably benefit enough with standard endocrine therapy; whereas, those with high Ki67 scores might need further adjuvant treatments [19]. A Ki67 > 10% after two to four weeks of endocrine therapy has been suggested as a cutoff for the early identification of nonresponders with an increased risk of relapse [20]. Similarly to other authors in various trials [37,38], we consider Ki67 > 10% after four weeks to be an indicator of innate endocrine resistance and, therefore, a criterion for withdrawing NET. For most of the patients whose Ki67 proliferation rates did not reduce, surgery was then indicated.

Most of the tumors in our cohort were T2. The maximum tumor diameter was reduced by 40%, which probably indicates a much larger volume decrease. This enabled BCS to be performed on over 85% of the patients; these are figures that are hardly possible in average conditions [39], and are slightly higher than other likely figures for BCS reported after NET [40,41], although they may be explained by a mean tumor size that was not excessively large. Downstaging in the axilla was not achieved, since all the cN1 tumors were at least pN1, and some cN0 tumors were pN1 after NET. In our cohort, ultrasound FNR was close to 25%, which is consistent with the rates published by other authors, who found higher FNR in luminal tumors [42–46]. On the other hand, axillary downstaging with NET is rather uncommon, with nodal pCR rates ranging from 1.3% to 3.0%, at least when it is applied over short periods (<6 months) [35,36,47], as was the case with most of our patients. As pointed out by both Hammond and Rusz et al., these numbers could be increased with longer periods of NET [37,48]. Axillary management after NET is not well defined in clinical guidelines. Most surgeons seem to apply the same criteria to NET as they do to NCT [49], but the context of NCT (mainly because it is usually indicated for high-risk tumors) does not seem applicable to NET, since any low-burden disease left in the axilla may not significantly impact the prognosis [50]. Kantor et al. analyzed the type of axillary surgery and residual nodal disease burden after NET in over 6500 patients, finding limited axillary disease (ypN1) in more than 90% of the cases [50]. In the study published by Weiss et al., including stage II–III ER+/HER2- patients, those receiving NET (2138) were more likely to be managed with an SLNB than those treated with NCT [51]. Based on this, we consider that the Z0011 criteria for avoiding axillary dissection can be safely applied to these patients, as they can to those who undergo upfront surgery. Therefore, only a small percentage of patients in our cohort needed an axillary dissection, and an SLNB was feasible in most cases.

In luminal-like breast cancers, pCR is infrequent with both NET and NCT. In the meta-analysis published by Spring et al., the pCR rates were <10% in both the NCT and NET arms [52]. In any case, the absence of pCR does not impact survival as much as with triple-negative or HER2+ tumors [22], meaning that it is not an optimal indicator of outcomes after NET. No cases of pCR were reported in our cohort. The PEPI score constitutes a validated prognostic model [5] that could be analogous to pCR, with escalation of therapy for those patients with PEPI > 0 and de-escalation (avoiding chemotherapy) for patients with a PEPI-0 status. In our cohort, almost half of the patients presented a PEPI score of zero. We also found that a larger Ki67 decrease after four weeks of treatment was related to a lower PEPI score, meaning that Ki67 changes indicate higher responsiveness to NET, with prognostic implications, and NET continuation should be discouraged when the Ki67 rate does not readily decline. One inconvenience of the PEPI score is that nodal involvement is a qualitative measurement, so a positive axilla always implies a score ≥3, regardless of whether one or more than three nodes are involved. Nevertheless, rigid parallelism between nodal involvement and chemotherapy should be avoided, as recently demonstrated in the

RxPONDER trial, where most of the ER+/HER2- tumors in postmenopausal women, with 1–3 nodes involved, had a low recurrence score and, therefore, would not have benefited from chemotherapy [53]. Consequently, the response to NET may be a helpful tool in guiding adjuvant treatments. Interestingly, the findings of the Z1031 trial show that tumors that do not lower Ki67 after two to four weeks also fail to respond well to chemotherapy, indicating intrinsic endocrine resistance, which makes it essential to consider alternative therapies [20]. Genomic profiling performed on diagnostic core biopsies can also provide useful information for predicting NET response or endocrine resistance, although larger prospective studies in this area are necessary in order to establish which group of genes can provide the maximum information, and, therefore, act as a reliable biomarker [3]. In our study, a poor response to NET, as well as other risk factors, such as large nodal involvement, were considered when indicating adjuvant chemotherapy, which was necessary in one fourth of our patients.

The value of our study is that it is based on a single-institution prospective cohort and the use, outside a clinical trial, of Ki67 reduction as a clinical tool for maintaining or withdrawing NET. An early decline in Ki67 levels is a sensitive biomarker of endocrine responsiveness, with remarkable prognosis implications. This allows NET to be offered safely to a broader range of patients, including those with luminal B-like features, provided that a reduction in the Ki67 proliferation rate is recorded. Nevertheless, we acknowledge that there are limitations in our study, because of the nonrandomization of patients, as well as the short-term follow-up. Only mild adverse effects were reported, and no treatment abandonment occurred, which is consistent with data from other studies [23].

In conclusion, the results from our study indicate that NET can be considered a well-tolerated and effective alternative for postmenopausal women with ER+/HER2- breast cancer, and that it increases the rates of BCS in both luminal A-like and luminal B-like tumors. Early changes in Ki67 proliferation rates become a feasible and reliable tool that provides crucial information about endocrine responsiveness and patient prognosis.

**Author Contributions:** Conceptualization, C.M. and J.I.S.-M.; methodology, C.M. and J.I.S.-M.; software, J.I.S.-M.; validation, C.M. and J.I.S.-M.; formal analysis, C.M. and J.I.S.-M.; investigation, C.M., L.Y., and J.I.S.-M.; resources, C.M., L.Y., J.M.O., E.M., L.F., A.B., A.L., M.M., M.J.R., V.C., M.Á.R., D.H., and J.I.S.-M.; data curation, C.M. and J.I.S.-M.; writing—original draft preparation, C.M. and J.I.S.-M.; writing—review and editing, C.M. and J.I.S.-M.; visualization, C.M. and J.I.S.-M.; supervision, C.M., D.H., and J.I.S.-M. All authors have read and agreed to the published version of the manuscript.

**Funding:** This research received no external funding.

**Institutional Review Board Statement:** The study was conducted in accordance with the Declaration of Helsinki and approved by the Ethics Committee of Hospital Universitario La Paz (protocol code PI-3031, approved 16 February 2018).

**Informed Consent Statement:** Informed consent was obtained from all subjects involved in the study.

**Data Availability Statement:** Not applicable.

**Acknowledgments:** The authors wish to thank Ralph Pemberton for his invaluable contribution to the translation and correction of this manuscript.

**Conflicts of Interest:** The authors declare no conflict of interest.

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
