# Peer review of "The Clinical Impact of Neoadjuvant Endocrine Treatment on Luminal-like Breast Cancers and Its Prognostic Significance: Results from a Single-Institution Prospective Cohort Study"

_curroncol, doi:10.3390/curroncol29040179_

Round 1

Reviewer 1 Report

Kudos to the authors on a timely, important manuscript summarizing the clinical impact of neoadjuvant endocrine therapy in postmenopausal patients with luminal A/B breast cancer. 

Overall, the study is well done, informative with sound statistical analysis and conclusions. However, authors can improve a few areas to make this manuscript even more appealing.

First, each table and figure can use more clarifying legends, expanding the content within each table so that the readers simply look at each figure or table, can grasp information immediately. in the current format, the readers have to go into the text and find relevant information about each content. it is always helpful if individual content can be understood on its own.

Second, the figure 3, the association between the Ki-67 fold change and PEPI and luminal A vs B - is informative. However, the way the result is presented can be improved. For instance, instead of simply mentioning Ki-67 fold change on either side, the Ki-67 fold change can be indicated on one of the sides, but change the subtitle of the item on the X horizon - to clarify what each parameter is. This feedback is in line with the first - that figure can be improved to be better understood.

Third, as a minor question - in figure 1, the good responders had either Ki-67 drop <10% or had clinically measured size reduction. was there any patient who dropped <10% but did not have any size reduction? Did both parameters need to be satisfied to move to a good responder?

I would also suggest - instead of good vs bad responders, 'responders vs non-responders. 

The discussion part is well done, however, a few sentences to suggest a future direction, indicating how these PEPI scores based on the NET with heterogeneous population/biological factor, including N node - may translate into the potential EFS, and how this can be better understood in the context of long-term follow up in adjuvant chemotherapy and survival data in luminal breast cancer, should improve the points of discussion.

Author Response

Dear reviewer,

First of all, thank you very much for taking your time for reviewing this manuscript. We appreciate all of your comments.

  1. Regarding tables, we have tried to make them more clear, especially table 1, which we have separated into 2 different.
  2. Following your suggestion, we have changed figure 3.
  3. When considering responders/no-responders (we have also changed that in figure 1, our protocol), we initially consider response only if a ki67 drop <10% after 4 wk is produced. In those patients with initial low Ki67 a reduction in size is considered. After that first evaluation, a clinical and ultrasound measurement is performed in order to consider response is going on. Figure 1 resumes this point.
  4. We think that the message in the discussion is that Ki67 changes are related to PEPI score, and both parameters are clearly associated to prognosis (as demonstrated by Ellis et al. in 2008 and in the very recent POETIC trial by Smith et al.). Therefore, when demonstrating that early Ki67  decrease, we can point out which patients will do better and which will not. In many of those cases with an endocrine response, chemotherapy may be avoided.

Thank you very much again for your comments. Best regards

Reviewer 2 Report

Dear Editor, thank you very much for the invitation to review this paper.

The manuscript titled “The clinical impact of neoadjuvant endocrine treatment on luminal-like breast cancer and its prognostic significance: results from a single-institution prospective cohort study” by Marti et al analyze a prospective cohort of patients treated with NET evaluating safety and effectiveness of NET in tumoral downstaging (size of tumors and/or Ki downstaging). This is a well-designed prospective single-institution study and encompasses patients seen from 2016 to 2020. As the authors should be aware, recent studies are using genomic and molecular platforms to select patients who are candidates for neoadjuvant endocrine therapy, especially in patients with proven axillary involvement at diagnosis. Moreover, there are ongoing studies testing the combination of cyclins with NET for the management of these patients (CORALLEEN, NEOPAL, PALLET, NeoMONARCH, FELINE, CARABELA…). Thus, I have several concerns that need to be addressed before considering publication. All these shortcomings must be addressed before this paper could be seriously considered for publication. 

 Minor changes: 

  1. Line 55-57: these statements are too restrictive, and the references provided, except for the last one, are too old. NET can produce a decrease in ki67 and a cell cycle arrest, but it is not clear that this translates into good long-term results (the phrase is too restrictive, on a topic that is not at all clear today). In more recent studies with cyclin + HT inhibitors in neoadjuvant therapy (NeoMonarch, feline...) it has been observed that the combination shows biological activity (significant decrease in Ki67 at 2 weeks). But it does not seem to improve the overall response rate or the pCR rate. So, I think this statement should be reconsidered.

  1. Line 72-75: The safety in terms of adherence and tolerability of endocrine therapy has been extensively studied in specific studies designed for this purpose. As the authors indicate, the study analyzes the effectiveness of NET, but I do not think that in this study a primary objective is to assess the safety of these drugs. In the results, the authors indicate that there were no patients who did not tolerate the treatment, but there are no data on the adverse effects of hormone therapy that usually occur in a percentage of users of hormone therapy. It could be said that the study assesses treatment tolerance in this population as a secondary objective (providing more information about it).

  1. Please consider checking the numbers in the tables as the sum does not always match the value 100%. Example:

o  Table 1: the sum of the percentages in clinical stage and in histological grade add up to 99.9%. It should be 100%.

  • Table 2: the sum of the percentages in pathological node status (62.7 + 38.3 = 101%) is not correct. It should be 100%.

Major changes: 

a. The authors conclude in the abstract that the rate of conservative surgery increases with the use of neoadjuvant endocrine therapy, since 85.2% of patients had received conservative surgery. However, the number of patients who were NOT candidates for conservative surgery before neoadjuvant treatment is unknown. Could you indicate the number of patients in which the indication changed from mastectomy to conservative surgery after neoadjuvant hormone therapy?

b. It would be important to know the performance status (ECOG, WHO) of the patients before starting treatment with hormone therapy. It is an important data to consider in a study whose median age of the patients is 69 years old. Do you have that data?

c. It should be clarified why 21 patients with axillary involvement before hormone therapy were included. Currently, there are ongoing and published studies indicating that the use of genomic tests and/or molecular portraits is an important element in selecting patients to receive this treatment (neoadjuvant endocrine therapy):

  1. Were they asked for any genomic test or molecular portrait?
  2. If yes, what was the result?
  3. Has any patient received adjuvant chemotherapy?
  4. It would be important to know the final pN (including the exact number of affected axillary nodes) at least in patients who received axillary lymphadenectomy.
  5. If genomic or molecular platform was not performed in the neoadjuvant setting, was performed in high-risk patients after surgery? If so, which results were obtained and if the subsequent management of the patient changed.

d. Although this is a study in which the last patient was included in 2020, it would be important to know if any patients had any loco-regional or distant relapse during the years of follow-up.

e. Could you please comment on what is the current treatment in your hospital for Luminal B cN+ patients? Do you consider that the management offered during the time of the study for these patients is still adequate? If not, should be included in the discussion. If only in selected patients, please indicate the selection criteria.

your sincerely

Author Response

Dear reviewer,

First of all, thank you very much for taking your time to review this manuscript. We appreciate all the comments and suggestions you make.

  1. Regarding Ki67 decrease, there is plenty of literature that relates it to a better prognosis. The POETIC trial (published by Smith et al. in Lancet Oncology 2020), found, with over 4,500 patients, that after a short treatment of 14 days before surgery with an aromatase inhibitor, those patients with initial low ki67 or with low ki67 after a 2week treatment period, had a better prognosis, almost 1/3 lower 5y recurrence risk when compared to those with no Ki67 decrease (21.5%). As the authors point out in the implications of the available evidence  "tumours with a high proliferation rate to derive information on early endocrine responsiveness that can be used to predict a patient’s 5-year prognosis on standard adjuvant therapy. The clinical manoeuvres to incorporate this in the patient pathway with reliable quality assured Ki67 are straightforward and the measurement of Ki67 is inexpensive, potentially making this an attractive approach to estimating the prognosis of patients with early breast cancer." A Ki67 low level after NET also contributes to a low PEPI score, as demonstrated already a few years ago by Ellis et al (Outcome prediction for oestrogen receptor positive breast cancer based on postneoadjuvant endocrine therapy tumor characteristics J Natl Cancer Inst 2008). On the other hand, pCR is not a good biomarker of response for luminal cancer as we remark in the manuscript, because these tumors do rarely respond completely either after chemo or endocrine therapy. Obviously there is a role for other therapies like cyclin inhibitors, which have demonstrated a higher cycle arrest (higher Ki67 decrease), although the clinical benefit seems meagre( NeoPalAna, PALLET,NeoMONARCH, CORALLEEN...). Maybe, these treatments will  be useful in neoadjuvance in high risk patients (high axillary burden, high grade...), but long-term survival data are required.
  2. We absolutely agree with you in that we cannot use the word safety as we do not provide data regarding side effects of the endocrine treatment. We have removed safety, and only refer to adherence and tolerance of the treatment. Thanks for this important appreciation.
  3. We have also reviewed the tables. There was an erratum in table 2, and the problem in table 1 is that, when using only one decimal number, sometimes the add doesn't go to 100% (we have recalculated so it gets there).
  4. Regarding how many patients would have not benefitted from breast conserving surgery, unfortunately, we don't have this data, although we agree with you that this would be the ideal scenario and should be reported when evaluating the patient for the first time. Anyway, most of the tumors were over 2cm and, although BCS may be feasible in some of these cases, it clearly increases the rate and simplicity of the surgery, as pointed out by many other authors. Using NET for decreasing tumor size is one of its main indications, as many clinical guidelines recommend.
  5. We do not routinely calculate de ECOG performance status, but all of the patients that go through surgery are at least an ECOG 0-2, otherwise we try to avoid surgery (at our facility we have a wide experience in tumor crioablation).
  6. Regarding nodal involvement, when we first designed the study we decided to include also N+ patients(mainly with low burden disease), as most of the NET clinical trials and studies do so, although nodal response is usually scarce. At that time, results from the Rxponder were not available and, nowadays, we are not allowed to perform a genomic profile in the core biopsy in most of the cases. Anyway, we consider it a very interesting option that may enhance choosing. between NET or NCT. A genomic profile (EP clin in our case) was performed in 12 patients either because axillary involvement or because a poor response to NET was recorded (generally a high Ki67 despite treatment). A high risk score was obtained in 5 patients, all of them received latter chemo, and low risk in 7 patients. In any case, treating luminal N+ cancers with NET for a period of time does not impact survival, moreover if a response to treatment is recorded (Johnston et al. Ann. Oncol. 2012, 23, 2296–2300; Pepping et al.Curr. Geriatr. Rep. 2017, 6, 239–246; Ailes et al. The American Surgeon 2021 0, 1-9).
  7. As explained in the manuscript (last paragraph of section 3.2 Treatment characteristics), 101 patients received radiotherapy and 29 patients, chemotherapy. Poor-no response to NET was taken into account when considering chemo.
  8. Of the 16 patients were an axillary dissection was performed, in 2 patients, no other nodes (besides SN) was positive. In 5, 1-2 nodes. And in the rest of them(8), more than 3 positive nodes were obtained. We did not find this information relevant as the aim of the study is to assess the role of Ki67 as an indicator of endocrine responsiveness. 
  9. Only one patient out of 115 had a relapse. It was an 85 year woman, that interrupted adjuvant endocrine therapy (apparently, not because an intolerance). She had a local relapse and died two years later. The other 114 patients, are free of disease and alive so far.
  10. At our facility, all breast cancers cases are discussed in the Tumor Board. If a luminal B-like cancer is presented, and the patient is postmenopausal, NET is considered as long as there is not a high axillary burden or a low axillary burden is combined with other features (like G3). In any case, we routinely perform a 4week biopsy to all patients undergoing NET and only continue it if Ki67 is below 10%.

In conclusion, we find ki67 changes a very useful tool to determine NET responsiveness. Nevertheless, combining this with a genomic profile may broaden its indication to some high risk patients, as demonstrated in the ADAPT study ("First results from the prospective high risk cohort from a large prospective phase III ADAPT trial provide evidence for good prognosis in some pts with >4 positive LN and e.g. low RS. Moreover combination of lower post-endocrine Ki-67 and limited tumor burden may be a promising criterion for CT de-escalation strategies even in patients with high RS"), presented at last SABCS.

Thank you very much for your comments. Best regards

Round 2

Reviewer 2 Report

Thanks to the authors for all the clarifications made. These works contribute to improving the care of our patients affected by breast cancer.